# Hypoxia-induced mitochondrial abnormalities in cells of the placenta

**Philippe Vangrieken** [1,2]*, **Salwan Al-Nasiry** [3], **Aalt Bast**[1], **Pieter A. Leermakers** [1], **Christy B. M. Tulen**[1], **Ger. M. J. Janssen**[4], **Iris Kaminski**[1], **Iris Geomini**[1], **Titus Lemmens**[1], **Paul M. H. Schiffers**[4], **Frederik J. van Schooten**[1], **Alex H. V. Remels**[1]

1 Department of Pharmacology and Toxicology, School of Nutrition and Translational Research in Metabolism (NUTRIM), Maastricht University Medical Center+, Maastricht, The Netherlands, 2 Department of Internal Medicine, School for Cardiovascular Diseases (CARIM), Maastricht University Medical Center+, Maastricht, The Netherlands, 3 Department of Obstetrics and Gynaecology, School for Oncology and Developmental Biology (GROW), Maastricht University Medical Center+, Maastricht, The Netherlands, 4 Department of Pharmacology and Toxicology, School for Cardiovascular Diseases (CARIM), Maastricht University Medical Center+, Maastricht, The Netherlands

* p.vangrieken@maastrichtuniversity.nl

## Abstract

### Introduction

Impaired utero-placental perfusion is a well-known feature of early preeclampsia and is associated with placental hypoxia and oxidative stress. Although aberrations at the level of the mitochondrion have been implicated in PE pathophysiology, whether or not hypoxia-induced mitochondrial abnormalities contribute to placental oxidative stress is unknown.

### Methods

We explored whether abnormalities in mitochondrial metabolism contribute to hypoxia-induced placental oxidative stress by using both healthy term placentae as well as a tropho-blast cell line (BeWo cells) exposed to hypoxia. Furthermore, we explored the therapeutic potential of the antioxidants MitoQ and quercetin in preventing hypoxia-induced placental oxidative stress.

### Results

Both in placental explants as well as BeWo cells, hypoxia resulted in reductions in mitochon-drial content, decreased abundance of key molecules involved in the electron transport chain and increased expression and activity of glycolytic enzymes. Furthermore, expression levels of key regulators of mitochondrial biogenesis were decreased while the abundance of constituents of the mitophagy, autophagy and mitochondrial fission machinery was increased in response to hypoxia. In addition, placental hypoxia was associated with increased oxidative stress, inflammation, and apoptosis. Moreover, experiments with MitoQ revealed that hypoxia-induced reactive oxygen species originated from the mitochondria in the trophoblasts.

**Data Availability Statement:** All relevant data are within the manuscript and its Supporting Information files.

**Funding:** This study was supported by NUTRIM Graduate Program. There was no additional external funding received for this study. The funders had no role in study design, data collection and analysis, decision to publish, or preparation of the manuscript. The first author (Philippe Vangrieken), received a salary by the NUTRIM Graduate Program.

**Competing interests:** The authors have declared that no competing interests exist.

**Abbreviations:** ABTS, 2,2'-azino-bis(3-ethylbenzothiazoline-6-sulphonic acid; BAX, Pro-apoptotic Bcl-2-associated X protein; BCL, Anti-apoptotic B-cell lymphoma 2; BNIP3, BCL2/adenovirus E1B 19 kDa protein-interacting protein 3; BNIP3L, BCL2/adenovirus E1B 19 kDa protein-interacting protein 3-like; Cat1, Catalase-1; COX, Cytochrome c oxidase; CS, Citrate synthase; DCFH-DA, 2',7'-dichlorodihydrofluorescein diacetate; DNM1L, Dynamin-related protein 1; ERR, Estrogen-related receptor; ETC, Electron transport chain; FAO, Fatty acid-β oxidation; Fis-1, Fission 1 protein; FUNDC1, FUN14 domain containing 1; GABARAPL1, gamma-aminobutyric acid type A receptor associated protein like 1; GLUT1, Glucose transporter 1; GSH, Glutathione; GSSG, Glutathione disulfide; HADH, β-hydroxyacyl-CoA dehydrogenase; HKII, Hexokinase; HRP, Horseradish peroxidase; IUGR, intrauterine growth restriction; ISSHP, International Society for the study of Hypertension in Pregnancy; LC3, Microtubule-associated protein 1 light chain 3; Mfn, Mitofusin; MnSOD2, Manganese-dependent superoxide dismutase; mtDNA, mitochondrial DNA; NRF, Nuclear respiratory factor; Opa1, Optic atrophy protein 1; OPTN, Optineurin; PARK2, E3 ubiquitin-protein ligase Parkin; PE, Preeclampsia; PFK, Phosphofructokinase; PGC-1, Peroxisome proliferator-activated receptor gamma coactivator 1; PINK1, PTEN-induced kinase 1; PPAR, Peroxisome proliferator-activated receptor; ROS, Reactive oxygen species; RPL13A, Ribosomal Protein L13a; SOD1, Superoxide dismutase 1; SQSTM1, Sequestosome 1; TEAC, Trolox Equivalent Antioxidant Capacity; Tfam, Transcription factor A; TNF-α, Tumor necrosis factor alpha; UQCRC2, Ubiquinol cytochrome c reductase core protein 2.

## Discussion

This study is the first to demonstrate that placental hypoxia is associated with mitochondrial-generated reactive oxygen species and significant alterations in the molecular pathways controlling mitochondrial content and function. Furthermore, our data indicate that targeting mitochondrial oxidative stress may have therapeutic benefit in the management of pathologies related to placental hypoxia.

## Introduction

Preeclampsia (PE) is a progressive multisystem pregnancy disorder that affects 2% to 8% of all pregnancies [1]. PE is mainly characterized by new onset of hypertension and proteinuria, caused by placental dysfunction. Although systemic pathological changes usually resolve following delivery of the placenta, it still predisposes mothers to cardiovascular, renal, and chorionic hypertensive disease in later life [1, 2]. Currently, there is no treatment and management of PE predominantly involves stabilization of the mother and fetus with timely delivery to prevent deterioration of the condition and subsequently limit morbidity and mortality as a result of the disorder [2].

Although the pathogenesis of PE remains poorly understood, the current paradigm revolves around oxidative stress in the trophoblast, the cell type that forms the epithelial lining of the placental villi that is in direct contact with the maternal blood [3, 4]. Oxidative stress has been shown to induce apoptosis in trophoblasts, accelerate trophoblast turn-over and promote excessive release of placental cytotoxic factors including e.g. pro-inflammatory cytokines and vasoactive compounds in the maternal circulatory system [5]. These cytotoxic factors subsequently drive PE-like symptoms including endothelial dysfunction, systemic hypertension and end-organ hypo-perfusion [5, 6]. Different stressors can lead to perturbation of placental endocrine actions of which the predominant one in PE is utero-placental malperfusion secondary to abnormal remodeling of the uterine spiral arteries [7, 8].

Disturbed delivery of blood, causing local hypoxia, is known to cause damage to the mitochondrial electron transport chain (ETC) which, as the ETC is the main site for intracellular oxygen consumption in the cell, can lead to increased production of reactive oxygen species (ROS) [2, 5]. Interestingly, recent evidence suggests abnormalities at the level of the mitochondrion in PE-complicated placentae [9]. However, whether or not hypoxia triggers these abnormalities and if this is linked to the development of oxidative stress in the placenta remains unclear.

Key mechanisms controlling mitochondrial function and content include biogenesis of new organelles (mitochondrial biogenesis), selective autophagy-mediated degradation of mitochondria (mitophagy) as well as mitochondrial fusion and fission events [10, 11]. Mitochondrial biogenesis is essentially controlled by the peroxisome proliferator-activated receptor (PPAR) gamma coactivator 1 (PGC-1) molecules that co-activate different nuclear proteins with transcriptional activity such as estrogen-related receptors (ERRs), PPARs, and nuclear respiratory factor 1/2α (NRF-1/2α), and mitochondrial transcription factor A (Tfam). These proteins collectively drive the expression of mitochondrial proteins and replication of mitochondrial DNA (mtDNA) [12]. Mitophagy can be divided into receptor-mediated and ubiquitin-mediated mitophagy. During receptor-mediated mitophagy, mitochondrial-based receptor proteins e.g. FUND14 domain containing 1 (FUNDC1), BCL2/adenovirus E1B 19 kDa protein-interacting protein 3 (BNIP3) or BNIP3 like (BNIP3L), are activated. Subsequently, these

activated receptor proteins recruit and bind a mammalian homologue of yeast autophagy-related protein 8, including microtubule-associated protein 1 light chain 3 beta (LC3B) and gamma-aminobutyric acid type A receptor associated protein like 1 (GABARAPL1) and so predestines the organelle for authophagic degradation. The ubiquitin-mediated mitophagy pathway is initiated by accumulation of PTEN induced putative kinase 1 (PINK1) on the mito-chondrial membrane, where they recruit E3 ubiquitin-protein ligase Parkin (PARK2) and/or several autophagy receptors (e.g. optineurin (OPTN)). When activated, PARK2 is able to ubi-quitinate several different mitochondrial substrates, including mitofusin 1 (Mfn1) and Mfn2, which in turn recruit the LIR-containing autophagy adaptor protein sequestome-1 (SQSTM1) and LC3B resulting in degradation of the organelle.

Unraveling the relationship between hypoxia, the regulatory pathways controlling mito-chondrial content and function, and oxidative stress in the trophoblast will enhance our insight into the complex pathogenesis of PE and may pave the way for new treatment modali-ties. Therefore, in this study, we exposed placental villous explants from non-complicated pregnancies as well as a trophoblast cell line (BeWo cells) to hypoxia and comprehensively assessed multiple indices of mitochondrial health, cellular oxidative stress, and antioxidant sys-tems. Parameters related to the molecular regulation of mitochondrial health included assess-ment of transcript and protein abundance as well as activity levels of constituents of mitochondrial metabolic pathways and key regulators of mitochondrial biogenesis, mitophagy, and the mitochondrial fusion and fission machinery. In addition, the potential of the flavonoid quercetin [13] as well as the mitochondrial-targeted antioxidant MitoQ to limit oxidative stress in BeWo cells was investigated.

## Materials and methods

### Clinical subject characteristics

Human term (>38 weeks gestational age) placentae were collected from cesarean deliveries performed by the Department of Obstetrics and Gynecology at the Maastricht University med-ical Center+. For this study, three placentae were used. Exclusion criteria were hypertension (pre-existing or onset during pregnancy) and/or proteinuria as defined by the International Society for the study of Hypertension in Pregnancy (ISSHP). The local medical ethics commit-tee Maastricht University Medical Center+ approved the procedures and all procedures on placentae were performed in accordance with the guidelines for experiments on human mate-rial (Maastricht University Medical Center+, Maastricht, the Netherlands; No. 16-4-047) by which placental samples were either collected as part of a prospective study for which partici-pants gave written and verbal consent or collected as "biological waste" from discarded tissue if participants did not verbally object to this happening, as documented by the health care pro-vider in their medical files.

### Villous explant isolation

As described earlier by Vangrieken *et al.* [14], three placental villous explants from non-com-plicated cesarean deliveries were processed directly after delivery. Villous explants were col-lected from the central region of the placentae at the maternal side. The basal plate of the specimens was removed and the remaining tissue was rinsed in a HEPES solution (NaCl 143.3 mM, KCl 4.7 mM, $MgSO_4$ 1.2 mM, $KH_2PO_4$ 1.2 mM, $CaCl_2$ 2.5 mM, glucose 5.5 mM, and HEPES 15 mM). Subsequently, per placenta, two villous explants of 30 g were collected and transferred into a 200 ml pre-warmed (37°C) HEPES buffer, kept in a warm water bath (37°C) and aerated under a standard culture condition (21% $O_2$) or a hypoxic condition (1% $O_2$) for 3

h. Tissue was collected and snap-frozen in liquid nitrogen, crushed by a mortar while frozen and stored at -80˚C until use.

## Cell culture

BeWo cells, a trophoblast cell line (ECACC, Porton Down, Salisbury, UK, No 86082803) were cultured according to the manufacture's protocol. Briefly, BeWo cells were cultured in Ham's F12 (Kaighn's) growth medium (Glutamine 2 mM and D-Glucose 7 mM, GIBCO, Carlsbad, CA, USA) enriched with 10% (vol/vol) fetal bovine serum (FBS) (SIGMA, St. Louis, MO, USA) and antibiotics (50 U/ml penicillin and 50 μg/ml streptomycin; both from GIBCO). Cells were routinely maintained in 175-$cm^2$ falcon flasks at pH 7.4 under 5% $CO_2$ and 95% humidity at 37˚C and passaged when reaching confluency of 70–80%. For enzyme activity assays and RNA/DNA isolation, 24 h after cell seeding in 6-wells plates ($3x10^5$ cells/well in 2 ml growth medium), cells were cultured for 24 h under a normoxic (control) condition (21% $O_2$, /5% $CO_2$) +/- quercetin (3 μM, Sigma, St. Louis, MO, USA) or MitoQ (mitoquinone mesylate) (1, 4 or 8 μM, MedChemExpress, New Jersey, USA) or a hypoxic condition (1% $O_2$ /5% $CO_2$) +/- quercetin (3 μM) or MitoQ (1, 4 or 8 μM). For detection of intracellular ROS levels, 72 h after cell seeding, cells were cultured for 24 h in black and clear bottom-96-wells plates (Costar/sigma Aldrich) ($1.5x10^4$ cells/well in 200 μl growth medium) under a normoxic (control) condition (21% $O_2$, /5% $CO_2$) +/- quercetin (3 μM) or MitoQ (1, 4 or 8 μM), or a hypoxic condition (1% $O_2$ /5% $CO_2$) +/- quercetin (3 μM), MitoQ (1, 4 or 8 μM) or $H_2O_2$ (100 μl, positive control).

## Villous explant and BeWo viability

Lactate dehydrogenase (LDH) activity, as an indicator of cell/tissue viability was measured in supernatant of both the cultured villous explants isolated from all 3 placentae and BeWo cells of the standard culture and the hypoxic condition using the cytotoxicity detection kit (Cat. No.11644793001, Roche, Mannheim, Germany).

## Preparation of the lysates

Approximately 40 mg of powdered placental tissue was homogenized for 10 sec at maximal speed with a hand-held PRO Scientific Bio-Gen PRO200 homogenizer in 800 μl KPE lysis buffer (13 mM $KH_2PO_4$, 68 mM $K_2HPO_4$, 9 mM EDTA, and 1% Triton X-100) for antioxidant enzyme activity assays or in 800 μl SET buffer (250 mM sucrose, 2 mM EDTA, 10 mM Tris, pH 7.4) for metabolic enzyme activity assays. For the generation of BeWo lysates, cells were washed twice with pre-cooled 1X Hank's balanced salt solution (HBSS), incubated for 15 min on ice in 500 μl KPE lysis buffer or in 100 μl SET buffer and collected using a rubber policeman. Subsequently, samples were homogenized for 10 sec at maximal speed with a handheld PRO Scientific Bio-Gen PRO200 homogenizer. KPE homogenates were centrifuged at 20,000 x *g* for 10 min at 4˚C. Subsequently, lysates (300 μl) were stored at -80˚C for the assessment of Trolox Equivalent Antioxidant Capacity (TEAC). The remaining lysate was mixed with 2.6% bovine serum albumin (BSA) (1:1) and stored at -80˚C for the determination of glutathione disulfide/glutathione (GSSG/GSH) levels. SET homogenates were snap-frozen in liquid nitrogen, defrosted, incubated on ice for 30 min and subsequently centrifuged at 20,000 x *g* for 2 min at 4˚C. 5% BSA was added to the lysate (1:4) and stored at -80˚C for the assessment of the citrate synthase (CS), β-hydroxyacyl-CoA dehydrogenase (HADH) and phosphofructokinase (PFK) activity. For preparation of the BeWo lysates for measuring L-Lactae, the manufacturer's protocol (L-Lactate Assay kit colometric, ab65331, Abcam, Cambridge, Massachusetts, USA) was followed. For DNA and RNA isolation, BeWo cells were washed twice with pre-

cooled 1X HBSS, incubated for 15 min on ice in 500 µl Trizol reagent and further processed according to the manufacturer's protocol (Catalog Number 15596026 and 15596018, Invitrogen™, USA). For generation of whole-cell lysates (for western blot analysis), BeWo cells were washed twice with pre-cooled 1X HBSS, incubated for 30 min on ice in 100 µl IP lysis buffer (50 mM Tris, 150 mM NaCl, 10% glycerol, 0.5% Nonidet P40, 1 mM EDTA, 1 mM $Na_3VO_4$, 5 mM NaF, 10 mM β-glycerophasphate, 1 mM $Na_4O_7P_2$, 1 mM DTT, 10 µg/µl leupeptin, 1% apropeptin, 1 mM PMSF, pH 7.4) and collected using a rubber policeman. Subsequently, samples were homogenized for 20 sec at maximal speed using the homogenizer. Lysates were incubated for 30 min on ice and centrifuged at 20,000 x $g$ for 30 min at 4˚C. Lysates were aliquoted (1 µg/µl) in Laemmli buffer (0.25 M Tris-HCl, 8% (w/v) SDS, 40% (v/v) glycerol, 0.4 M DTT, 0.04% (w/v) Bromphenol Blue, pH 6.8) and boiled for 5 min at 95˚C. Protein concentrations of whole-cell lysates and enzyme activity lysates were determined using the Pierce™ BCA Protein Assay kit according to the manufacturer's protocol (Pierce Chemical Co., Rockford, IL).

## Citrate synthase (CS) activity

As previously described (CS; EC 2.3.3.1) [15], a reaction mix was set up in 96 well plates in duplicate containing 5 µl undiluted sample, 200 µl reagent containing Tris (100 mM), DNTB (0.1 mM) and acetyl-coenzym A (0.3 mM). The reaction was started with 5 µl start reagent containing oxaloacetic acid (25 mM). Enzyme activity was monitored at 412 nm (37˚C) and corrected for total protein concentration.

## β-hydroxyacyl-CoA dehydrogenase (HADH) activity

As previously described (HADH; EC 1.1.1.35) [16], a reaction mix was set up in a 96 well plate in duplicate containing 10 µl undiluted sample, 100 µl reagent containing NADH (1.1 mM), tetrapotassium pyrophosphate (100 mM). The reaction was started with 10 µl acetyl-coenzym A (2.4 mM). Enzyme activity was kinetically-monitored at 340 nm (37˚C) and corrected for total protein concentration.

## Phosphofructokinase (PFK) activity

As previously described ((PFK, EC 2.7.1.11) [17], a reaction mix was set up in a 96 well plate in duplicate containing 20 µl undiluted sample, 100 µl reagent containing Tris Base (49.6 mM), $MgCl_2.6H_2O$ (7.4 mM), KCl (3.2 mM), KCN (384.6 µM), ATP (3.0 mM), DTT (1.5 mM), NADH (0.3 mM), aldolase (0.019 U), glycerol-3-phosphate dehydrogenase (0.019 U) and triose phosphate, isomerase (0.019 U), pH 8.0. The reaction was started with fructose-6-phosphate (35.9 mM) in Tris buffer (49.6 mM), pH 8.0. Enzyme activity was monitored at 340 nm (37˚C) and corrected for total protein concentration.

## L-Lactate assay

As instructed by the manufacturer's protocol (L-Lactate Assay kit colometric, ab65331, Abcam, Cambridge, Massachusetts, USA), the reaction was started by adding reaction mix to the sample wells and incubated for 30 minutes. L-Lactate was measured at 450 nm (37˚C).

## Quantitative real-time PCR

cDNA was synthesized from 400 ng RNA using the iScript™ cDNA synthesis kit (Bio-Rad, Laboratories, Hercules, CA, USA) according to the manufacturer's protocol. 4.4 µl of 1:50 diluted cDNA was used for quantitative PCR amplification using target-specific primers (S1 Table) and 2X Sensimix™ SYBER® & Fluorescein mix (Bioline, Alphen aan de Rijn, the

Netherlands) on a LightCycler480 384-wells PCR machine (Roche, Almere, the Netherlands). Specificity of PCR amplification was checked by melt curve analysis. Expression levels of genes of interest were corrected using normalization factor calculated based on the expression of 2 different housekeeping genes (*Cyclophilin A* and *RPL13A*), which were found to be most stable from a selection of 3 genes by using the GeNorm software (Primerdesign, Southamton, USA). The list of primers can be found in S1 Table.

## Mitochondrial DNA (mtDNA) copy number

An amount of 4.4 μl of 1:25 diluted DNA was used for qPCR as described above, using mitochondrial DNA (mtDNA, Cytochrome *c* oxidase subunit II (*COXII*)) and genomic DNA (gDNA, Ribosomal Protein L13a (*RPL13A*))–specific primers (S1 Table). mtDNA/gDNA ratio was determined by dividing the relative quantity of mtDNA by the relative quantity of gDNA.

## JC-10 mitochondrial membrane potential

Mitochondrial membrane potential was assessed as instructed by the manufacturer's protocol (JC-10 assay kit, ab112134, Abcam, Cambridge, Massachusetts, USA) according to the manufacturer's protocol. Briefly, BeWo cells were plated overnight at 40,000 cells/well in a 96-well plate and subsequently cultured under a normoxic (control) condition (21% $O_2$, /5% $CO_2$), or a hypoxic condition (1% $O_2$ /5% $CO_2$) for 24 h. Then JC-10 dye-loading solution was added in incubated for 30 min (37˚C) and assay buffer was added. The fluorescent intensities for both J-aggregates and monometric forms JC-10 were measured at 515 nm and 570 nm (37˚C).

## Western blotting

As described earlier by Leermakers *et al*. [18], 10 μg of protein was run through a Criterion XT 4–12 or 12% Bis-Tris gel (Bio-Rad, Veenendaal, the Netherlands) in 1x MES running buffer (Bio-Rad, Veenendaal, the Netherlands) at 100 volts, and was subsequently blotted on a Nitrocellulose membrane (Bio-Rad Laboratories B.V., Veenendaal, the Netherlands) by electroblotting. At least two protein ladders were loaded on each gel (Precision Plus Protein™ All Blue Standards #161–0373, Bio-Rad Laboratories, Veenendaal, the Netherlands). Membranes were stained with 0.2% PonceauS in 1% acetic acid (Sigma-Aldrich, Zwijndrecht, the Netherlands) for 5 min, washed with milliQ and imaged using the Amersham™ Imager 600 (GE Healthcare, Eindhoven, the Netherlands) to quantify total protein content as a correction for gel-loading. Membranes were blocked for 1 h with Tween20 Tris-buffered saline (TBST; 20 mM Tris, 137 mM NaCl, 0.1% (vol/vol) Tween20, pH 7.6) containing 3% (w/v) non-fat dry milk (Campina, Eindhoven, the Netherlands), washed, and incubated overnight at 4˚C with a target-specific primary antibody (Table 1) diluted 1:1,000–1:10,000 in TBST with 3% (w/v) BSA or non-fat dry milk at 4˚C. Subsequently, membranes were washed and incubated with a HRP-conjugated secondary antibody (#BA-9200, #BA-1000, Vector Laboratories, Amsterdam, the Netherlands), diluted 1:10,000 in 3% (w/v) non-fat dry milk in TBST for 1 h at room temperature. Membranes were then washed, incubated for 5 min with 0.5x SuperSignal West PICO or 0.25x West Femto Chemiluminescent Substrate (Thermo Scientific, Landsmeer, the Netherlands) and imaged using the Amersham™ Imager 600. Original unaltered images were quantified with Image Quant software (GE Healthcare, Eindhoven, the Netherlands). Measured protein quantity was corrected for total protein content. Images included in the figures of this manuscript have been adjusted for brightness and contrast equally throughout the picture.

**Table 1. Antibodies used for western blot.**

| Target | RRID | Company | Product number | Dilution |
|---|---|---|---|---|
| HKII | AB_2232946 | Cell Signaling Technology | Cat# 2867 | 1:1000 |
| OXPHOS | AB_2629281 | MitoScience LLC | Cat# MS604 | 1:1000 |
| PGC-1α | AB_10697773 | Millipore | Cat# 516557 | 1:1000 |
| NRF1 | AB_2154534 | Abcam | Cat# ab55744 | 1:1000 |
| ERRα | AB_1523580 | Abcam | Cat# ab76228 | 1:1000 |
| Tfam | AB_10682431 | Millipore | Cat# DR1071 | 1:1000 |
| SQSTM1 | AB_10624872 | Cell Signaling Technology | Cat# 5114 | 1:1000 |
| PINK1 | AB_10127658 | Novus Biologicals | Cat# BC100-494 | 1:2000 |
| PARK2 | AB_2159920 | Cell Signaling Technology | Cat# 4211 | 1:1000 |
| FUNDC1 | AB_10609242 | Santa Cruz Biotechnology | Cat# sc-133597 | 1:500 |
| BNIP3 | AB_2259284 | Cell Signaling Technology | Cat# 3769S | 1:1000 |
| BNIP3L | AB_2688036 | Cell Signaling Technology | Cat# 12396 | 1:1000 |
| GABARAPL1 | AB_2294415 | Proteintech Group | Cat# 11010-1-AP | 1:1000 |
| LC3B | AB_915950 | Cell Signaling Technology | Cat# 2775 | 1:1000 |
| DNM1L | AB_10950498 | Cell Signaling Technology | Cat# 8570 | 1:1000 |
| Caspase-3 | AB_2341188 | Cell Signaling Technology | Cat# 9661 | 1:1000 |

HKII: Hexokinase II, OXPHOS: Oxidative phosphorylation, antibody cocktail (containing NADH: Ubiquinone oxidoreductase subunit B8 (NDUFB8), Succinate dehydrogenase complex, subunit B (SDHB), ubiquinol cytochrome c reductase core protein 2 (UQCRC2), Mitochondrially encoded cytochrome c oxidase I (mt-COI), ATP synthase, H+ transporting, mitochondrial F1 complex, alpha (ATP5A)), PGC-1α: Proliferative activated receptor gamma, coactivator 1 alpha, NRF1: Nuclear respiratory factor 1, ERRα: Estrogen Related Receptor alpha, Tfam: Mitochondrial transcription factor A, SQSTM1: Sequestosome 1, PINK1: PTEN-induced kinase 1, PARK2: Parkin, FUNDC1: FUN14 domain-containing protein 1, BNIP3: BCL2/Adenovirus E1B 19 kDa protein-interacting protein 3, GABARAPL1: γ-aminobutiric acid receptor-associated protein-like 1, LC3B: Microtubule associated protein 1A/1B-light chain 3 beta and DNM1L: Dynamin 1 Like.

## Glutathione disulfide/glutathione (GSSG/GSH) levels

The GSH-assay was performed for the determination of the levels of GSH + GSSG and GSSG as described previously [21]. GSH (0.1–10 µM) and GSSG (0.1–5 µM) standards were prepared in KPE buffer and 1.3% 5-sulfosalicylic acid. GSSG standards and samples were diluted 1:10 with neat 2-vinylpyridine, incubated and mixed for 1 h to form a stable complex with GSH, preventing it from participating in the enzymatic recycling reaction with glutathione reductase. Reactions were set up in a 96 well plate and 50 µl of the sample was loaded in duplicate. Reactions were initiated by adding 100 µl 0.8 mM NADPH/0.6 mM DTNB 1:1 and 4 U/ml GSSG reductase to the samples. Color development of the samples and standards was recorded kinetically for 3 min in 9 reads at 412 nM resulting in GSH + GSSG and GSH slope values. Activity was corrected for protein content of the samples and expressed in nM/mg protein/min.

## Trolox Equivalent Antioxidant Capacity (TEAC)

The Trolox equivalent antioxidant capacity (TEAC value) is a measurement for total antioxidant status, relating the free radical scavenging properties of a solution or a compound to that of the synthetic antioxidant Trolox and was performed as described earlier [19]. Briefly, a 5 mM 2,2'-azino-bis(3-ethylbenzothiazoline-6-sulphonic acid) (ABTS) solution was prepared in 145 mM sodium phosphate buffer (pH 7.4). Then, an ABTS$^{\bullet-}$ solution was prepared by adding 10 µl 1/100 horseradish peroxidase (HRP) and 10 µl of 2 mM $H_2O_2$ solution and diluted in an ABTS solution to a final absorbance of 0.70 ± 0.02 at 734 nm at 37°C. Deproteinization of the samples was performed by adding 10% trichloroacetic acid (TCA) (1:1) to the samples. For

measuring antioxidant capacity, 50 μl of the lysate was mixed with 950 μl ABTS$^{\bullet-}$ solution at 37°C for 5 min and absorbance was measured at 734 nm and compared to the absorbance of an ABTS$^{\bullet-}$ solution without sample. Absorbance was corrected for total protein content.

### Intracellular ROS levels

As previously described [20], intracellular ROS levels were quantified using the 2',7'-dichloro-dihydrofluorescein diacetate (DCFH-DA)-assay. 72 h after cell seeding, cells were washed twice with 150 μl HBSS, incubated for 1 h with 100 μl H$_2$DCFH-DA (50 μM) and washed twice again with 150 μl HBSS. After 24 h culture under the different conditions in duplicate as described above, DCF was measured at λ excitation = 485 nm and λ emission = 525 nm and corrected for total protein content.

### Statistical analysis

Data are depicted as bar graphs indicating the mean and SEM as fold change compared to the control. For each comparison, the D'agrostino and Pearson omnibus normality test was used to test normality and subsequently either an unpaired t-test or Mann Whitney test was used accordingly (GraphPad Software, La Jolla California USA). A p-value <0.05 was considered as significantly different from the control group and was presented as follows: Ns: p >0.05, *p ≤0.05, **p ≤0.01, ***p ≤0.001. For each assay, three independent experiments were performed in which each experimental condition contained 3–12 biological replicates. For subsequent data analysis, replicates were expressed as fold change compared to the control and were subsequently pooled.

## Results

Placental villous explants were exposed to hypoxia for up to 3 h since previous studies using placental villous explants *ex vivo* showed that the viability of first and third-trimester tissue in culture could only be maintained for 4 h [21]. Longer exposure times lead to pronounced tissue deterioration and massive apoptosis of the trophoblasts in placental villous explants indicating that, in this model, it is only suitable to study acute effects [22]. To study the effects of long-term exposure to hypoxia and to determine if changes are trophoblast-specific, BeWo cells were used.

### Villous explant and BeWo viability

No significant differences were found for LDH activity between the control- and hypoxic-conditioned villous explant medium indicating that viability of the placental villous explants was comparable between the two conditions after 3 h of incubation. Furthermore, when BeWo cells were cultured for 6 h, 24 h and 48 h under a normoxic or hypoxic condition, only at 48 h a significant increase in LDH activity was observed in cell supernatant of BeWo cells exposed to hypoxia compared to the control condition (S2 Fig). Since cell viability in the hypoxic condition significantly decreased after 48 h culturing under hypoxia, compared to the control condition, and therefore affecting the readouts of our study, 24 h exposure to hypoxia was used in this study.

### Increased oxidative stress levels in placental villous explants and BeWo cells upon hypoxia

We first determined the impact of hypoxia (1% O$_2$) on oxidative stress levels and known cellular antioxidant defense systems in placental villous explants as well as in cultured BeWo cells.

## Placental explants

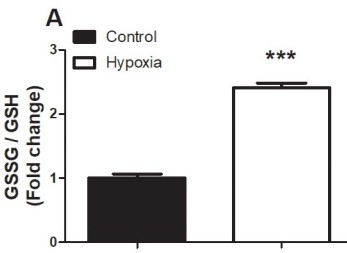
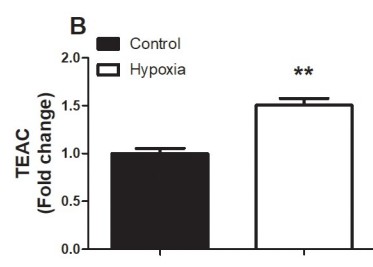
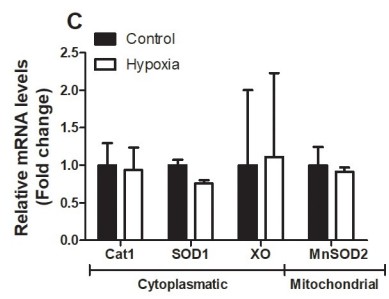

## BeWo cells

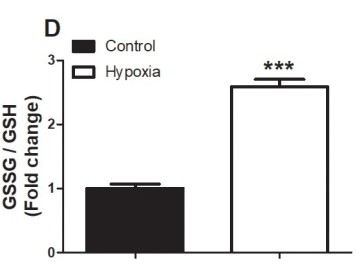
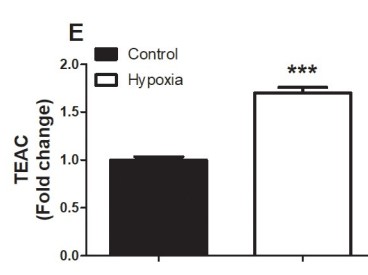
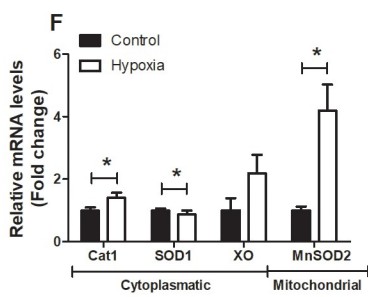

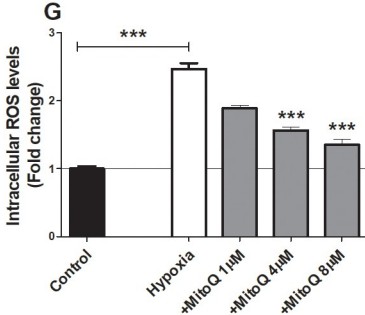
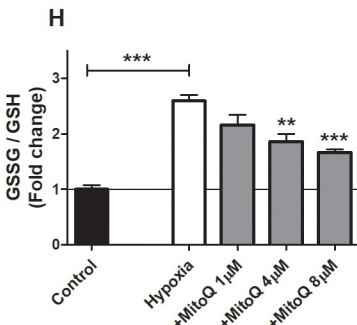

**Fig 1. Increased oxidative stress levels in placentae and trophoblasts upon hypoxia.** GSSG/GSH ratio (**A**), TEAC (**B**) and transcript levels of *CAT1*, *SOD1*, *XO* and *MnSOD2* (**C**) assessed in placental villous explants exposed to normoxia (control) or hypoxia for 3 h (n = 3) and GSSG/GSH ratio (**D**), TEAC (**E**) and transcript levels of *CAT1*, *SOD1*, *XO* and *MnSOD2* (**F**) assessed in trophoblasts exposed to hypoxia or normoxia (control) for 24h. Intracellular ROS levels using DCFH-DA (**G**) and GSSG/GSH ratio (**H**) were assessed in trophoblasts exposed to normoxia (control), hypoxia or hypoxia + MitoQ (1, 4 or 8 μM) (DCFH-DA-assay: n = 6-12/experimental condition (n = 3 experiments), GSSG/GSH-assay: n = 2–6 experimental condition (n = 3 experiments)). Data are presented as fold change compared to the control and as mean with SEM. $^*p \leq 0.05$, $^{**}p \leq 0.01$ and $^{***}p \leq 0.001$. GSSG: Oxidized glutathione, GSH: Reduced glutathione, TEAC: Trolox equivalent antioxidant capacity, Cat1: Catalase-1, SOD1: Superoxide dismutase 1, XO: Xanthine oxidase, MnSOD2: Manganese-dependent superoxide dismutase and DCFH-DA: 2',7'-Dichlorodihydrofluorescein diacetate.

As depicted in Fig 1, both the glutathione disulfide/glutathione (GSSG/GSH) ratio as well as the trolox equivalent antioxidant capacity (TEAC) increased significantly in both placental villous explants and BeWo cells exposed to hypoxia (3 h and 24 h respectively). Also, although placental villous explants exposed to hypoxia did not show alterations in mRNA abundance of known antioxidant enzymes, in BeWo cells exposed to hypoxia mRNA transcript levels of catalase 1 (*Cat1*) and mitochondrial manganese-dependent superoxide dismutase 2 (*MnSOD2*)

significantly increased while transcript levels of superoxide dismutase 1 (*SOD1*) significantly decreased (Fig 1A–1F).

As mitochondria are the main site for cellular oxygen consumption and the main source of cellular ROS production, we next investigated whether or not hypoxia-induced changes in ROS levels and antioxidant status originated from the mitochondria. Therefore, we used the mitochondrial-targeted antioxidant MitoQ in cultured BeWo cells during exposure to hypoxia. Intracellular ROS levels, as well as oxidized glutathione levels increased more than two-fold in BeWo cells exposed to hypoxia compared to normoxia-exposed BeWo cells, and MitoQ normalized both hypoxia-induced increases in intracellular ROS formation and oxidized glutathione levels in BeWo cells in a concentration-dependent manner (Fig 1G and 1H). In addition, as shown in S3 Fig, the antioxidant quercetin did not reduce hypoxia-induced intracellular ROS formation. Interestingly, the combination of the lowest concentration of MitoQ (1 μM) and quercetin (3 μM) significantly decreased hypoxia-induced intracellular ROS production and levels of oxidized glutathione to levels similar as in the normoxia-exposed BeWo cells (Fig 1G and 1H).

In addition to these indicators of oxidative stress, mRNA transcript levels of tumor necrosis factor alpha (*TNF-α*), interleukin 6 (*IL-6*) and interleukin 8 (*IL-8*) as well as the pro-apoptotic mRNA transcript ratio of Bcl-2associated X protein / anti-apoptotic B-cell lymphoma 2 (*BAX/BCL-2*) and protein levels of Cleaved Caspase-3 (17 + 15 KDa), were all significantly increased in BeWo cells in response to 24 h of hypoxia but were unaltered in placental villous explants after a 3 h exposure to hypoxia (S4A–S4E Fig).

## Hypoxia-induced decrease in mitochondrial content in placental villous explants and BeWo cells

Having established that hypoxia exposure in BeWo cells, results in mitochondrial ROS production and considering the fact that ROS can damage mtDNA [23], we next assessed indices of mitochondrial content in placental villous explants as well as in cultured BeWo cells exposed to hypoxia. As shown in Fig 2, hypoxia-induced a profound reduction in mitochondrial content in both placental villous explants as well as in cultured BeWo cells. This was evidenced by a reduction in mtDNA copy number and a decrease in the enzyme activity of citrate synthase (CS) (Fig 2A–2D), two well-described indicators of mitochondrial content [24]. Moreover, the combination MitoQ and quercetin, in concentrations that completely normalized hypoxia-induced ROS formation (S3A and S3B Fig), ameliorated hypoxia-induced reductions in mtDNA copy number in BeWo cells (Fig 2E). Subsequently, mitochondrial membrane potential (ΔΨm) was assessed in BeWo cells stained with JC-10, after exposure to hypoxia. JC-10 is selectively taken up into mitochondria and reversibly changes color from green to orange with increased membrane potential. Hence, decreased mitochondrial depolarization was demonstrated by a decrease in 570/515 fluorescence intensity ratios. Hypoxia-exposed BeWo cells show a significantly lower mitochondrial membrane potential compare to the control condition (Fig 2F) which is in line with the increased proapoptotic status as indicated in S4D Fig.

## Increased abundance and activity of glycolytic constituents in placental cells in response to hypoxia

Next, we investigated whether the decrease in mitochondrial content observed in placental villous explants and in BeWo cells exposed to hypoxia was associated with changes in (mitochondrial- and non-mitochondrial) metabolic processes. Therefore, we assessed the abundance and activity levels of key constituents of the ETC, fatty acid-β oxidation (FAO) and glycolysis. mRNA abundance as well as enzymatic activity of β-hydroxyacyl-CoA dehydrogenase (HADH), a key enzyme involved in mitochondrial FAO, were unaltered in response to

## Placental explants

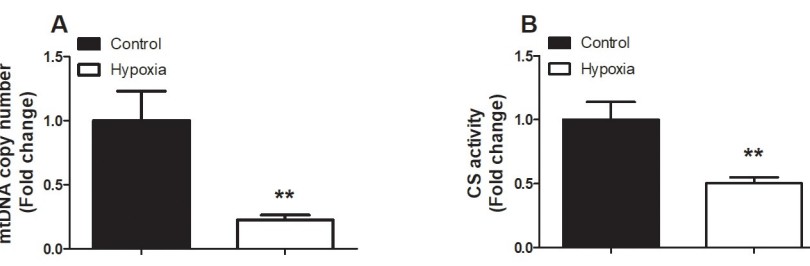

## BeWo cells

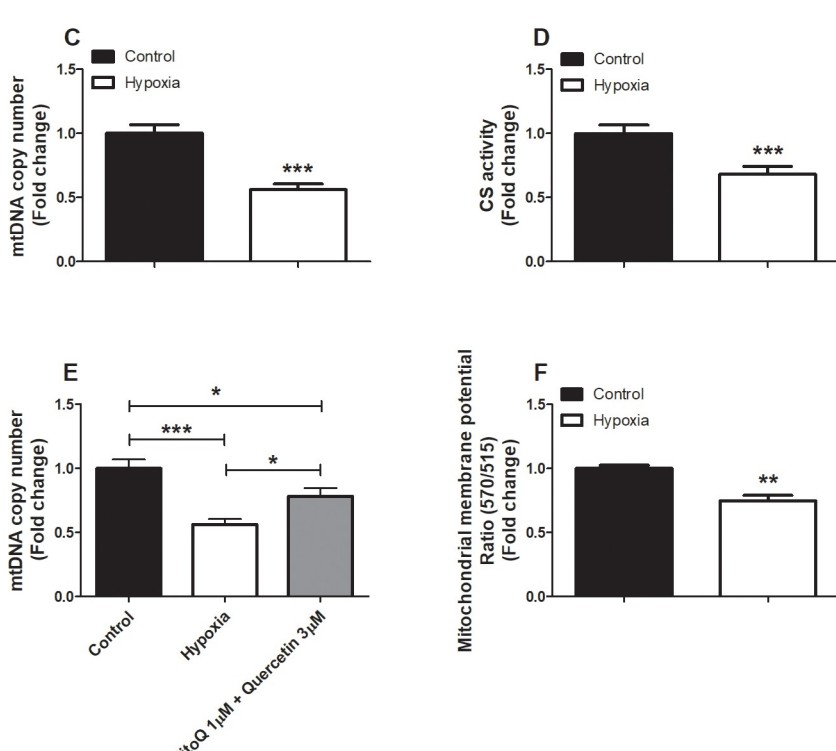

**Fig 2. Lower mitochondrial content in placental villous explants and trophoblasts upon hypoxia.** mtDNA copy number (A) and CS activity (B) assessed in placental villous explants exposed to hypoxia or normoxia (control) for 3 h (n = 3 in duplicate) and mtDNA copy number (C) and CS activity (D) assessed in trophoblasts exposed to normoxia (control) or hypoxia for 24 h. mtDNA copy number (E) was assessed in trophoblasts exposed to normoxia (control), hypoxia, hypoxia + MitoQ (1, 4 or 8 μM), quercetin (3 μM) or MitoQ (1 μM) + quercetin (3 μM) for 24 h (n = 3-6/ experimental condition (n = 3 experiments)) and mitochondrial membrane potential changes (F) were measured with JC-10 in BeWo cells exposed to hypoxia for 24 h (n = 3 experimental condition (3 experiments)). Data are presented as fold change compared to the control and as mean with SEM. $^*p \leq 0.05$, $^{**}p \leq 0.01$ and $^{***}p \leq 0.001$. mtDNA: Mitochondrial DNA and CS: Citrate synthase.

hypoxia in both placental villous explants as well as in BeWo cells (S5A–S5D Fig). Interestingly, while in placental villous explants exposed to hypoxia for 3 h transcript and protein abundance of both nuclear- or mitochondrial-encoded sub-units of ETC complexes were unaltered (S6A–S6C Fig), protein levels of ubiquinol cytochrome c reductase core protein 2

(UQCRC2, complex III) as well as mRNA transcript levels of all investigated sub-units, including mitochondrial-encoded *COXII*, were significantly decreased in BeWo cells after 24 h exposure to hypoxia (S6D–S6F Fig).

When assessing activity levels of phosphofructokinase (PFK), a key enzyme involved in glycolysis, we observed that hypoxia resulted in a significant increase in PFK activity in both placental villous explants as well as in BeWo cells. Moreover, in BeWo cells, but not placental villous explants, this was associated with a significant increase in protein and mRNA transcript levels of hexokinase II (*HKII*) as well as transcript levels of the glucose transporter 1 (*GLUT1*) (Fig 3A–3F). In line, exposing BeWo cells to hypoxia significantly increased L-lactate levels (Fig 3G).

## Placental explants

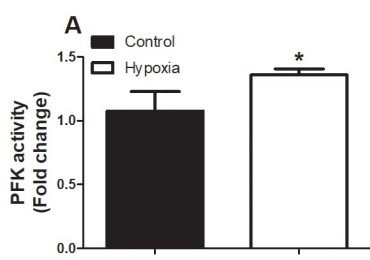
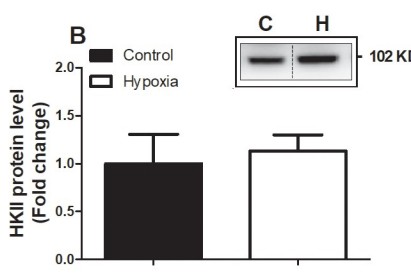
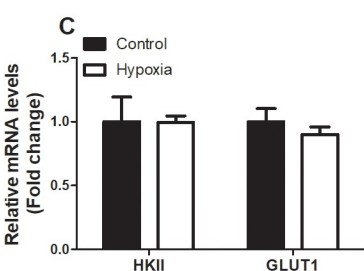

## BeWo cells

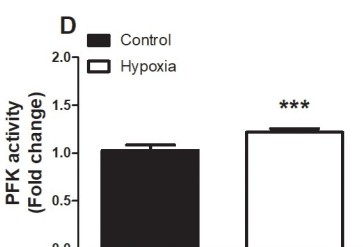
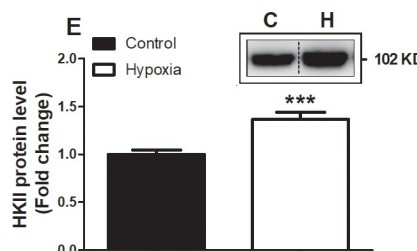
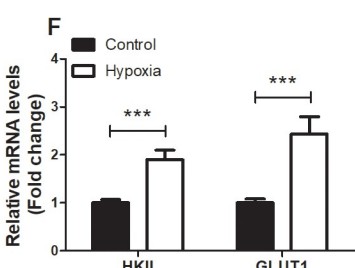

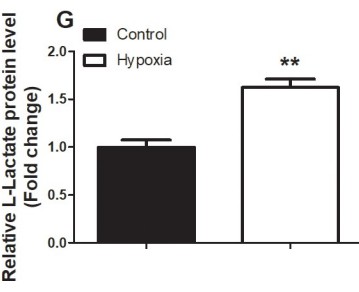

**Fig 3. Increased expression and activity of key glycolytic enzymes in placentae and trophoblasts upon hypoxia.** PFK enzyme activity (**A**), HKII protein level (**B**), mRNA expression levels of *HKII* and *Glut-1* (**C**), assessed in placental villous explants exposed to normoxia (control) or hypoxia for 3 h (n = 3 in duplicate (PFK assay) or (n = 3 (protein and mRNA)) and PFK enzyme activity (**D**), HKII protein level (**E**), mRNA expression levels of *HKII* and *Glut-1* (**F**) and L-Lactate (**G**), assessed in trophoblasts exposed to normoxia (control) or hypoxia for 24 h (n = 3-6/experimental condition (n = 3 experiments)). Representative immunoblots are shown and Western blots were corrected for total protein loading assessed by Ponceau S Staining with adjusted contrast equally applied to the whole photograph. Black boxes around the representative pictures indicate that they were cut from the same Western blot. Data are presented as fold change compared to the control and as mean with SEM. *p ≤ 0.05 and ***p ≤ 0.001. PFK: Phosphofructokinase, HKII: Hexokinase and GLUT1: Glucose transporter 1.

## Alterations in the molecular regulation of mitochondrial biogenesis in BeWo cells upon hypoxia

Considering the reductions in mitochondrial content in placental cells upon exposure to hypoxia, we next explored whether hypoxia affected the molecular regulation of mitochondrial biogenesis by the PGC-1 signaling network. As shown in Fig 4, while no differences were observed in the abundance of key regulators of mitochondrial biogenesis in placental villous explants exposed to hypoxia for 3h (Fig 4A and 4B), protein levels of Tfam and ERRα were significantly decreased in response to 24 h of hypoxia in BeWo cells. Furthermore, mRNA transcript levels of *PGC-1β*, *NRF2α*, *Tfam*, and *ERRα* were significantly decreased in BeWo cells upon 24 h of exposure to hypoxia (Fig 4C and 4D).

## Expression of key constituents of the mitophagy machinery is altered in BeWo cells upon hypoxia

As mitochondrial content is determined by the balance between mitochondrial biogenesis and breakdown through mitophagy (e.g. mitochondrial-specific autophagy), we next assessed the

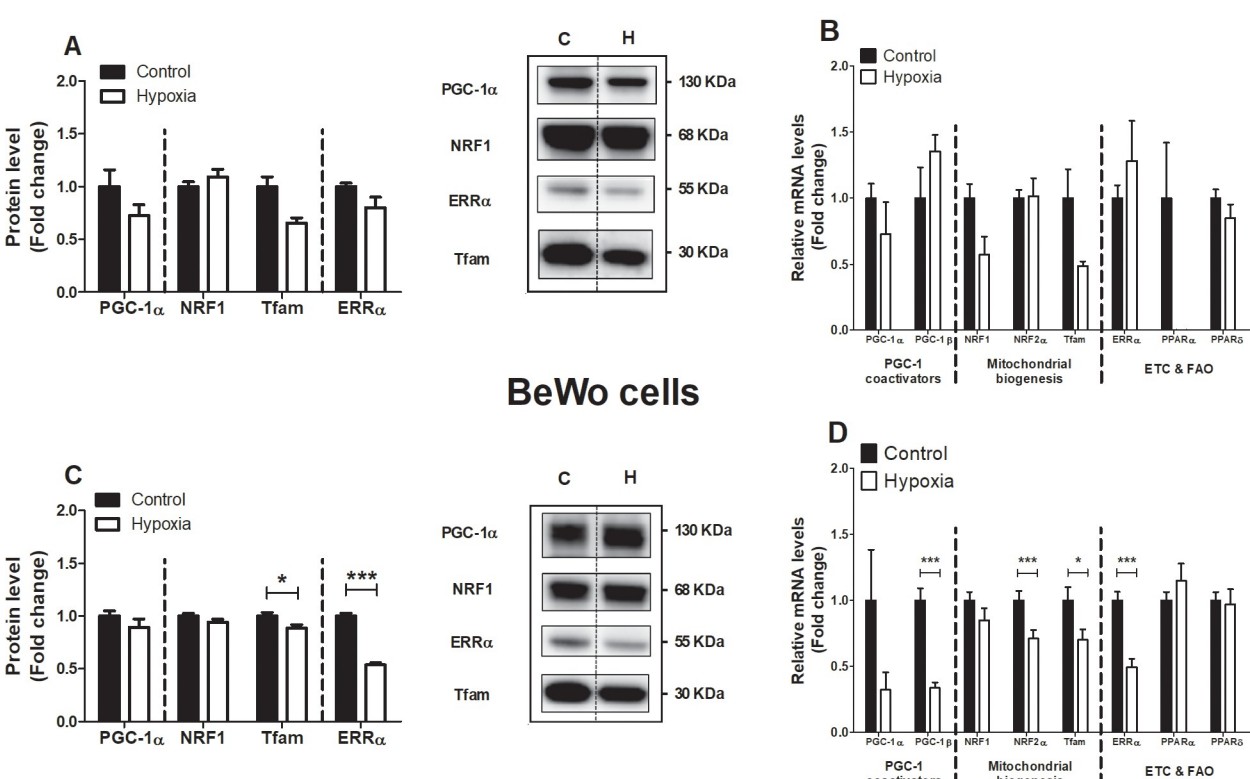

**Fig 4. Alterations in the molecular regulation of mitochondrial biogenesis in trophoblasts upon hypoxia.** Protein levels of PGC-1α, NRF1, Tfam and ERRα (**A**) and mRNA transcript levels of *PGC-1α*, *PGC-1β*, *NRF1*, NRF2α, *Tfam*, *ERRα*, *PPARα and PPARδ* (**B**) assessed in placental villous explants exposed to normoxia (control) or hypoxia for 3 h (n = 3) and protein levels of PGC-1α, NRF1, Tfam and ERRα (**C**) and mRNA transcript levels of *PGC-1α*, *PGC-1β*, *NRF1*, NRF2α, *Tfam*, *ERRα*, *PPARα and PPARδ* (**D**) assessed in trophoblasts exposed to normoxia (control) or hypoxia for 24h (n = 3-6/experimental condition (n = 3 experiments)). Representative immunoblots are shown and Western blots were corrected for total protein loading assessed by Ponceau S Staining with adjusted contrast equally applied to the whole photograph. Black boxes around the representative pictures indicate that they were cut from the same Western blot. Data are presented as fold change compared to the control and as mean with SEM. *$p \leq 0.05$ and ***$p \leq 0.001$. PGC-1α: Peroxisome proliferator-activated receptor gamma coactivator 1-alpha, NRF1: Nuclear respiratory factor 1, Tfam: Mitochondrial transcription factor A, ERRα: Estrogen-related receptor alpha, PGC-1β: Peroxisome proliferator-activated receptor gamma coactivator 1-beta, NRF2α: Nuclear respiratory factor 2 alpha, PPARα: Peroxisome proliferator-activated receptor alpha, PPARδ: Peroxisome proliferator-activated receptor delta.

## Placental explants

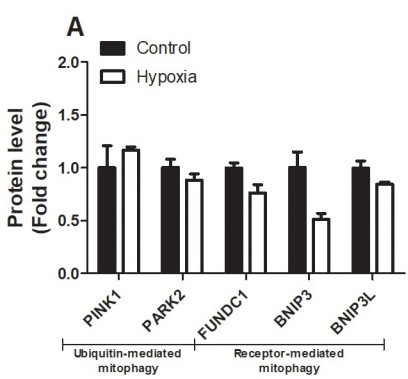
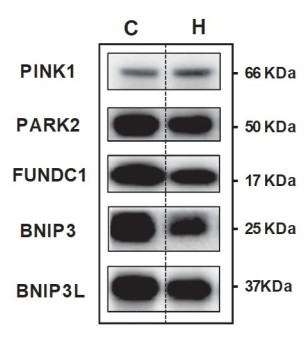
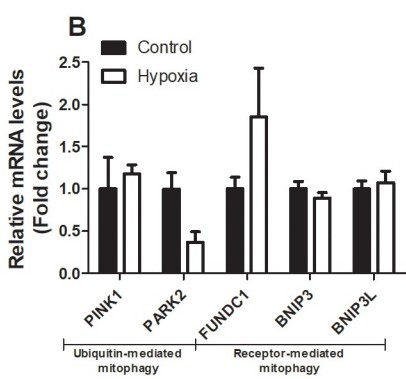

## BeWo cells

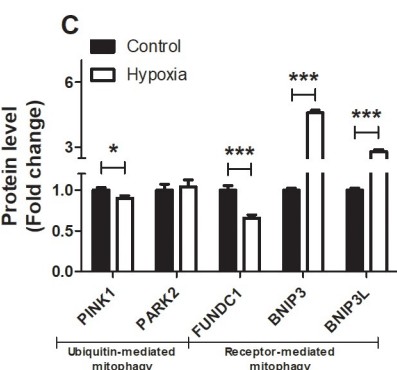
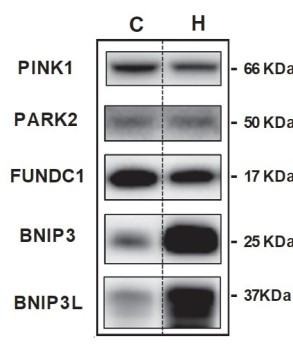
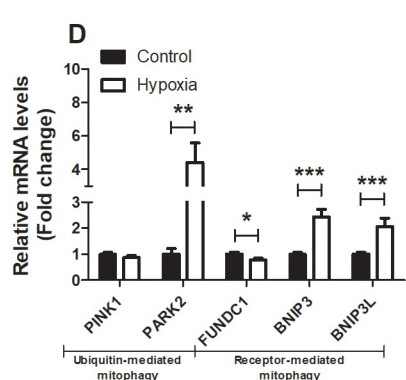

**Fig 5. Expression of key constituents of the mitophagy machinery is altered in trophoblasts upon hypoxia.** Mitophagy-associated protein levels of PINK1, PARK2, FUNDC1, BNIP3 and BNIP3L (**A**) and mRNA transcript levels of *PINK1*, *PARK2*, *FUNDC1*, *BNIP3* and *BNIP3L* (**B**) assessed in placental villous explants exposed to normoxia (control) or hypoxia for 3 h (n = 3) and mitophagy-associated protein levels of PINK1, PARK2, FUNDC1, BNIP3 and BNIP3L (**C**) and mRNA transcript levels of *PINK1*, *PARK2*, *FUNDC1*, *BNIP3* and *BNIP3L* (**D**) assessed in trophoblasts exposed to normoxia (control) or hypoxia for 24 h (n = 3-6/experimental condition (n = 3 experiments)). Representative immunoblots are shown and Western blots were corrected for total protein loading assessed by Ponceau S Staining with adjusted contrast equally applied to the whole photograph. Black boxes around the representative pictures indicate that they were cut from the same Western blot. Data are presented as fold change compared to the control and as mean with SEM. $^*p \leq 0.05$, $^{**}p \leq 0.01$ and $^{***}p \leq 0.001$. PINK1: PTEN-induced kinase 1, PARK2: E3 ubiquitin-protein ligase Parkin, FUNDC1: FUN14 domain containing 1, BNIP3: BCL2/adenovirus E1B 19 kDa protein-interacting protein 3 and BNIP3L: BCL2/adenovirus E1B 19 kDa protein-interacting protein 3-like.

impact of hypoxia on key constituents of the mitophagy machinery in placental villous explants and BeWo cells. As mitophagy requires several general autophagy-related proteins for generating the autophagosomal membrane and priming the autophagosome to the mitochondria, these proteins were studied as well.

All investigated constituents of the autophagy- (S7A and S7B Fig) and mitophagy machinery (Fig 5A and 5B) were unaltered in placental villous explants exposed for 3h to hypoxia compared to the control condition. In contrast, in BeWo cells exposed for 24 h to hypoxia, mitophagy-related protein levels of PINK1 and FUNDC1, as well as mRNA transcript levels of *FUNDC1*, were significantly decreased in response to hypoxia while both protein and mRNA transcript levels of *BNIP3* and *BNIP3L* were potently increased. In addition, while PARK2 protein levels were not affected in BeWo cells exposed to hypoxia, mRNA transcript levels of *PARK2* were significantly increased in response to hypoxia (Fig 5C and 5D). Also, in cultured BeWo cells, 24 h of hypoxia resulted in a significant decrease in autophagy-associated LC3BI

and LC3BII protein levels while the ratio of LC3BII/LC3BI was significantly increased compared to the normoxic condition. In addition, while both protein as well as mRNA levels of *GABARAPL* were increased in response to hypoxia, protein levels of SQSTM1 were decreased and its transcript levels increased upon hypoxia (S7C and S7D Fig). *OPTN* mRNA levels increased 2-fold in response to hypoxia (S7D Fig).

## Alterations of key constituents of mitochondrial fission and fusion in BeWo cells upon hypoxia

As mitochondrial fission and fusion are key events in the processes of mitophagy and mitochondrial biogenesis and are essential in maintaining normal mitochondrial homeostasis, mRNA and protein levels of mitochondrial fission and fusion proteins were investigated both in placental villous explants as well as BeWo cells exposed to hypoxia. As depicted in Fig 6A and 6B, no alterations were found in fission and fusion-related proteins and mRNA transcript

## Placental explants

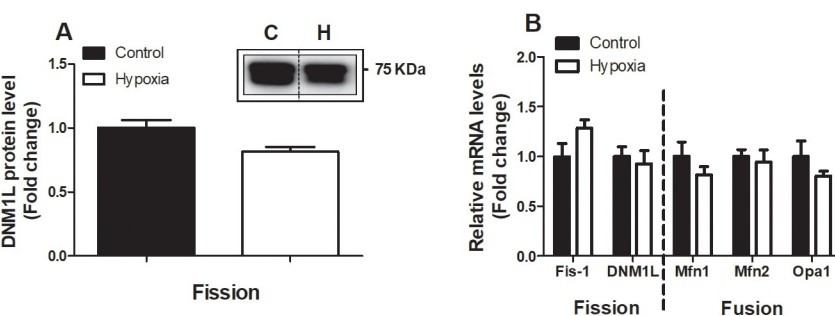

## BeWo cells

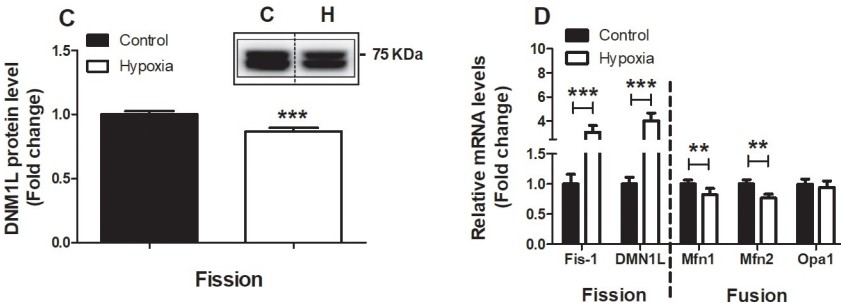

**Fig 6. Alterations of key constituents of mitochondrial fission and fusion in trophoblasts upon hypoxia.** Fission-related protein levels of DNM1L (**A**), fission-related mRNA transcript levels of *Fis-1* and *DNM1L* and fusion-related mRNA transcript levels of *Mfn1*, *Mfn2* and *Opa1* (**B**) assessed in placentae exposed to normoxia (control) or hypoxia for 3 h (n = 3) and fission-related protein levels of DNM1L (**C**), fission-related mRNA transcript levels of *Fis-1* and *DNM1L* and fusion-related mRNA transcript levels of *Mfn1*, *Mfn2* and *Opa1* (**D**) assessed in trophoblasts exposed to normoxia (control) or hypoxia for 24 h (n = 3-6/experimental condition (n = 3 experiments)). Representative immunoblots are shown and Western blots were corrected for total protein loading assessed by Ponceau S Staining with adjusted contrast equally applied to the whole photograph. Black boxes around the representative pictures indicate that they were cut from the same Western blot. Data are presented as fold change compared to the control and as mean with SEM. $^{**}p \leq 0.01$ and $^{***}p \leq 0.001$. DNM1L: Dynamin-1-like protein, Fis-1: Fission 1 protein, Mfn1: Mitofusin-1, Mfn2: Mitofusin-2 and Opa1: Optic atrophy protein 1.

levels in placental villous explants exposed for 3 h to hypoxia. In BeWo cells, fission related protein levels of dynamin-1-like protein (DNM1L), a key protein involved in mitochondrial fission [25], was significantly decreased after 24 h exposure to hypoxia. In addition, fission-related mRNA transcript levels of fission 1 (*Fis1*) and *DNM1L* were significantly increased after exposure to hypoxia, while mRNA expression of fusion-related transcript levels of mito-fusin-1 (*Mfn1*) and *Mfn2* was reduced in BeWo cells upon exposure to hypoxia (Fig 6C and 6D).

## Discussion

In the current study, we show that hypoxia triggers placental oxidative stress, which likely resulted from ROS formation and alterations in antioxidant status in trophoblasts. Moreover, we show for the first time that, in BeWo cells exposed to hypoxia, mitochondria are a major source of ROS and that hypoxia triggers a profound loss of mitochondrial content. Further-more, in response to hypoxia, the abundance and activity of key constituents of glycolysis rap-idly increased which, upon prolonged exposure to hypoxia in BeWo cells, was also associated with significant reductions in the abundance of subunits of the electron transport chain and changes in the expression of key regulatory molecules controlling mitochondrial biogenesis, mitophagy and mitochondrial fission and fusion events.

### Hypoxia-induced oxidative stress

The presence of placental oxidative stress and the link with PE pathophysiology is well-estab-lished. Indeed, there is abundant evidence that placental oxidative stress triggers the release of cytotoxic and vasoactive factors as well as pro-inflammatory cytokines into the maternal circu-lation driving PE symptoms like maternal endothelial dysfunction and hypertension [2]. One of the key triggers of placental oxidative stress that has been proposed in PE is hypoxia [5, 26]. In support of this notion, stabilization of hypoxia-inducible factor 1-alpha (HIF-1α) has been reported in PE and hypoxia-exposed placentae [27–29]. Furthermore, both in hypoxia-exposed trophoblasts as well as in a prenatal hypoxia rat model for PE increased oxidative stress has been reported [5, 26, 30]. In line with these observations, the GSSG/GSH ratio (Fig 1A and 1D) and intracellular ROS levels in our study were significantly increased in placental cells in response to hypoxia (Fig 1G and 1H). Furthermore, hypoxia markedly increased total antioxidant capacity and oxidized GSH both in placental explants as well as BeWo cells in our study (Fig 1A, 1B, 1D and 1E), which may be a compensatory response to increased oxidative stress [31, 32]. The fact that mitochondrial superoxide concentration is tightly controlled by the mitochondrial antioxidant $MnSOD_2$ [33], combined with the notion that 24 h exposure of BeWo cells to hypoxia in our study resulted in a 4-fold increase in transcript levels of *MnSOD2* (Fig 1F) suggests mitochondria to be a significant source of ROS. In contrast to observations in BeWo cells, despite the increased GSSG/GSH ratio in placental explants, no increase in MnSOD2 was observed. This may be explained by the short exposure of placental explants to hypoxia (3 h), compared to the 24 h exposure of BeWo cells to hypoxia. In agreement with the notion that hypoxia-induced oxidative stress may originate from the mitochondria, the mito-chondrial-targeted antioxidant MitoQ concentration-dependently reduced intracellular ROS levels and lowered the GSSG/GSH ratio in BeWo cells during a 24 h exposure to hypoxia in our study (Fig 1G and 1H). This is in line with previous finding by Aljunaidy *et al.* [26], who showed not only normalization of oxidative stress levels by MitoQ in a hypoxic rat model of reduced placental perfusion, but also protection against sex- and age-dependent alterations in cardiac and vascular health of offspring later in life.

## Hypoxia-induced inflammation and apoptosis

In addition to leading to oxidative stress, 24 h exposure to hypoxia in our study resulted in increased expression of inflammatory mediators in BeWo cells (S4C Fig) as well as an increased *BAX/BCL-2* mRNA expression ratio and increased protein levels of Cleaved Caspase-3 (S4D and S4E Fig), indicative of activation of apoptosis, which is in line with previous findings [34–36]. In agreement with previous studies, 3 h exposure to hypoxia, did not induce a significant alteration in the expression of inflammatory cytokines and the induction of apoptosis [37]. Furthermore, in response to hypoxia, our data showed decreased mitochondrial membrane potential in BeWo cells (Fig 2F), which is well-known to trigger apoptosis [38] and is also suggestive of a hypoxia-induced proapoptotic condition [39]. In line with this, it is well-established that both the induction of inflammation as well as apoptosis of placental cells are induced by hypoxia and are implicated in PE pathophysiology [40]. Indeed, women with PE show higher plasma levels of TNF-α and elevated TNF-α protein and mRNA levels in their placentae compared to normal pregnant women, [33]. Furthermore, TNF-α is known to be a potent stimulus for endothelin 1 secretion, which is known to be a major vasoactive compound released by the placenta under hypoxia, contributing to maternal hypertension [14, 41, 42]. With regard to apoptosis in our study, an increased *BAX/BCL-2* ratio BeWo cells exposed to hypoxia was also associated with an increased abundance of BNIP3 proteins (Fig 5C and 5D), which act as an activator for hypoxia-induced cell apoptosis and autophagy by triggering the BAX/Bak or LC-3/Beclin 1 signaling pathway [43, 44]. Several Bcl-2 family members including BAX, have been shown to be directly regulated by a lack of oxygen via stabilization of HIF-1 [45].

## Hypoxia-induced reduction in mitochondrial content

Interestingly, besides their traditional role in cellular energy metabolism, mitochondria are also important mediators of oxidative stress, inflammation, and apoptosis. Moreover, evidence is emerging that PE is associated with abnormalities at the levels of the mitochondrion in the placenta [46, 47]. Indeed, besides alterations in placental mitochondrial content in PE [48], several human studies showed significant reductions in adenosine triphosphate (ATP) levels in PE placentae, which is suggestive of impaired mitochondrial functioning [49–51]. Indicative of a role for hypoxia in mediating these changes, exposure of mice to lower oxygen (13% instead of 21% $O_2$) led to an overall reduction in placental mitochondrial oxidative capacity [52] which could be beneficial to keep $O_2$ transfer to the fetus constant [53]. This is in line with our findings as we reported evidence for significant reductions in mitochondrial content in both healthy placentae and BeWo cells exposed to hypoxia (Fig 2A–2E).

## Hypoxia-induced shift from oxidative to glycolytic energy metabolism

Furthermore, expression of complex III of the ETC, which is the major source of mitochondrial production ROS in hypoxic conditions [54], as well as transcript levels of complex I and IV including mitochondrial-encoded COXII were decreased in BeWo cells exposed to hypoxia (S6D–S6F Fig), which is in concert with earlier studies [55, 56]. Despite a similar overall trend that could be observed in placental explants exposed to hypoxia for 3 h, the relatively low power and short exposure time, compared to our *in vitro* experiments may explain the less pronounced alterations in gene and protein expression of the OXPHOS subunits (S6A–S6C Fig). In addition, 24 h exposure to hypoxia in BeWo cells was associated with increased key glycolytic enzyme activity and production of L-lactate in our study (Fig 3), which together with the above-described data are suggestive for a shift from oxidative to glycolytic energy metabolism. Exposing healthy placental explants to hypoxia for 3 h, significantly increased

PFK activity, which the most prominent regulator enzyme in glycolysis. Expression of *HKII* and *GLUT*, however, was not yet affected after 3 h exposure to hypoxia, indicating a dominant role of PFK in the hypoxia-induced induction of glycolysis, which earlier was already found to be hypoxia-sensitive as it is HIF-1-dependent [57]. In agreement with these findings, exposure to low oxygen led to an increased trans-placental transfer of glucose and amino acids in mice [58]. Moreover, oxidative stress during hypoxia is known to activate HIF which is known to increase transcription of glycolytic enzymes and promote anaerobic glycolysis to maintain ATP synthesis under low oxygen conditions [59]. In accordance with our results, showing increased transcription of GLUT1 upon 24 h exposure to hypoxia (Fig 3F), GLUT1 is known to be activated upon binding of HIF-1α to the hypoxia-responsive element in the GLUT-1 pro-motor, which has been observed earlier in both hypoxia-exposed and PE placentae [5, 60]. Col-lectively, this data in concert with available literature suggest that in response to hypoxia, BeWo cells quickly develop mitochondrial-generated oxidative stress, display a subsequent increase in glycolysis and by a decrease in expression of key components of the ETC. The increased reliance on glycolysis found in cells of PE placentae or upon hypoxia may be a conse-quence of the decreased mitochondrial content and impaired functioning of the ETC and thus the capacity for producing ATP. This shift to anaerobic respiration may also be a protective response of the placenta to the impaired placental perfusion in PE and so sparing $O_2$ for the growing fetus [53]. It is thus expected that increased glycolysis is more likely a consequence than involved in the onset of PE.

## Hypoxia-induced increase in mitophagy and fission

Reductions in oxidative respiration in hypoxic tissues have been associated with decreased mitochondrial biogenesis [61–63]. In line with this and with decreased mitochondrial content in placental cells in response to hypoxia that we observed, we also found significant reductions in expression levels of key regulators of mitochondrial biogenesis (Fig 4C and 4D), which is in accordance with recent studies [44]. The fact that we observed reduced mRNA levels of nuclear- as well as mitochondrial-encoded OXPHOS subunits in BeWo cells exposed to hyp-oxia for 24 h (S6E and S6F Fig) does suggest an impairment in the activity of mitochondrial biogenesis. Moreover, the expression of OXPHOS genes is tightly controlled by the same key regulators of mitochondrial biogenesis that we measured. More specifically, Tfam and ERRα, have been convincingly demonstrated to control expression of mitochondrial-encoded and nuclear-encoded OXPHOS genes respectively, which further strengthens our suggestion of hypoxia-induced impairment of mitochondrial biogenesis. The discrepancy (as can be observed in Fig 4) between changes in the protein and mRNA abundance of key regulators of mitochondrial biogenesis in response to hypoxia in placental explants and BeWo cells (although similar trends can be observed in both models) may be due to the duration of the hypoxic incubation and the relative contribution of the different cell types present in these models. Besides mitochondrial biogenesis, selective autophagy of mitochondria (i.e. mito-phagy) is heavily involved in the regulation of mitochondrial content in cells. Enhanced mito-phagy is suggested to be an adaptive response to reduce the levels of ROS and protect cell integrity during hypoxia. It has previously been found that 1% $O_2$ can lower mitochondrial mass up to 75% through the initiation of the HIF1-dependent expression of BNIP3, a mito-chondrial protein that competes with Bcl2 and thereby freeing beclin1 to trigger autophagy [64]. In our study, BeWo cells exposed to hypoxia showed an increased abundance of recep-tor-mediated mitophagy-related proteins BNIP3 and BNIP3L (Fig 5C and 5D), which is in line with findings in previous studies [44, 65, 66]. In line with the literature, hypoxia-induced increases in protein levels of BNIP3 and BNIP3L in BeWo cells were indeed associated with an

increased abundance of general autophagy-related GABARAPL1 and an increased ratio of LC3BII/LC3BI protein levels (S7C and S7D Fig), which are known to predispose mitochondria for degradation [67, 68]. In addition, abundance of receptor-mediated mitophagy-related protein FUNDC1 was significantly decreased in BeWo cells in response to hypoxia (Fig 5C and 5D), which is in line with the hypoxia-mediated degradation of FUNDC1 as described by Chen *et al.* [69]. Likely, these changes require some time to occur, as 3h of hypoxic incubation of placental explants did not reveal similar changes (Fig 5C and 5D). Collectively, these data suggest that the molecular control of receptor-mediated mitophagy in placentae is altered upon hypoxia, which may well contribute to the loss of mitochondrial content that we observed. In addition, mitochondrial biogenesis and mitophagy are highly integrated with mitochondrial fusion and fission events. Increased mitochondrial fission has been implicated in the pathogenesis of several diseases and is often linked with the increased presence of oxidative stress and apoptosis [70–73]. In this regard, increased levels of fission-related *Fis-1* and *DMN1L* transcripts and decreased levels of fusion-related *Mfn1* and *Mfn2* transcripts found in our study (Fig 6C and 6D), indicate that the balance between fusion and fission may be tilted towards increased mitochondrial fission in response to hypoxia in our study. It is important to highlight that the changes in the expression levels of these proteins/transcripts upon prolonged exposure to hypoxia, have been shown to be associated with activation of mitophagy [74, 75], are merely suggestive of changes in the molecular control of mitophagy. As a follow-up step, in order to confirm active mitophagy in these cells in response to hypoxia, it is suggested, to visualize mitochondria with electron microscopy to further morphometrically identify this process in more detail.

Collectively, with regard to the time-course of hypoxia-induced changes, our data show that oxidative stress in cells of the placenta develops rapidly in response to hypoxia, which is associated with profound reductions in mitochondrial content and increased activity of glycolytic enzymes. Alterations in the molecular mechanisms controlling mitochondrial content and function (mitochondrial biogenesis *vs* mitophagy), however, only become apparent after prolonged exposure to hypoxia (24h).

Interestingly, although MitoQ and quercetin together initiated complete normalization of intracellular ROS formation during hypoxia, MitoQ alone did not completely prevent intracellular ROS formation suggesting that other cellular ROS-generating systems contribute to hypoxia-induced oxidative stress in placental cells. In agreement with this, xanthine oxidase has been shown to be implicated in PE pathophysiology and has been suggested to contribute to (hypoxia-induced) placental oxidative stress [76, 77]. To further investigate whether these anti-oxidants may confer protection in villous explants *ex vivo*, it is suggested that this intervention regime is performed in (perfused) placental explants to further consolidate our findings in BeWo cells.

In conclusion, our results show hypoxia-induced abnormalities at the level of the mitochondrion and the molecular pathways controlling mitochondrial content and function in placentae and oxidative stress, which may contribute to PE pathogenesis or other hypoxia-related pregnancy disorders like intrauterine growth restriction (IUGR). Although the concept of hypoxia-induced mitochondrial abnormalities in cells of the placenta is not new [78], our study is the first to show that hypoxia-induced oxidative stress mainly originates from the mitochondria and is unique in comprehensively assessing not only indices of mitochondrial content but also multiple constituents of mitochondrial biogenesis and mitophagy in cells of the placenta in response to hypoxia.

## Supporting information

**S1 Fig. Blot/Gel image data of all conditions including Ponceau S staining.** No significant differences were found for LDH activity between the control- and hypoxic-conditioned villous

explant medium indicating that viability of the placental villous explants was comparable between the two conditions after 3 h of incubation. Furthermore, when BeWo cells were cultured for 6 h, 24 h and 48 h under a normoxic or hypoxic condition, only at 48 h a significant increase in LDH activity was observed in cell supernatant of BeWo cells exposed to hypoxia compared to the control condition (S2 Fig). Since cell viability in the hypoxic condition significantly decreased after 48 h culturing under hypoxia, compared to the control condition, and therefore affecting the readouts of our study, 24 h exposure to hypoxia was used in this study. (PDF)

**S2 Fig. Villous explant and BeWo viability.** LDH activity between the control- and hypoxic-conditioned villous explant after 3 h of incubation (**A**) and in BeWo cells after 6 h, 24 h and 48 h incubation (**B**). To confirm whether mitochondria are a significant source for ROS in placentae upon hypoxia, the effect of the mitochondrially targeted antioxidant MitoQ and the systemically acting antioxidant quercetin on ROS formation and oxidative status was tested. Intracellular ROS levels increased more than two-fold in trophoblasts exposed for 24 h to hypoxia compared to normoxia-exposed trophoblasts (S3A Fig). Moreover, the mitochondrially targeted antioxidant MitoQ (4 and 8 μM) ameliorated hypoxia-induced intracellular ROS formation in trophoblasts in a concentration-dependent manner. The systemically acting antioxidant quercetin (3 μM), did not induced significant reductions in intracellular ROS formation during exposure to hypoxia. Interestingly, the combination of the lowest concentration of MitoQ (1 μM) and quercetin (3 μM) significantly decreased intracellular ROS production upon 24 h exposure to hypoxia and was the only condition that was not significantly different from the normoxic condition (S3A Fig). In agreement, the hypoxia-induced increase of the GSSG/GSH ratio, could be normalized by MitoQ (4 and 8 μM) and quercetin (3 μM), but were still significantly different compared to the control condition (S3B Fig). Interestingly, the combination of the lowest concentration of MitoQ (1 μM) and quercetin (3 μM), normalized the GSSG/GSH ratio during exposure to hypoxia (S3B Fig).
(TIF)

**S3 Fig. Antioxidant intervention prevented hypoxia-induced oxidative stress in trophoblasts.** Intracellular ROS levels using DCFH-DA (**A**) and GSSG/GSH ratio (**B**) were assessed in trophoblasts exposed to normoxia (control), hypoxia, hypoxia + MitoQ (1, 4 or 8 μM), quercetin (3 μM) or MitoQ (1 μM) + quercetin (3 μM) for 3 h (DCFH-DA-assay: n = 6-12/experimental condition (n = 3 experiments) and GSSG/GSH-assay: n = 6, 2 and 2/experimental condition (n = 3 experiments)). Data are presented as fold change compared to the control and as mean with SEM. *p ≤ 0.05, **p ≤ 0.01 and ***p ≤ 0.001. DCFH-DA: 2',7'-Dichlorodihydrofluorescein diacetate and GSSG: Glutathione disulfide, GSH: Glutathione. While placentae exposed to hypoxia did not show alterations in proinflammatory and apoptotic-related mRNA transcripts, mRNA levels of tumor necrosis factor-α (*TNF-α*), interleukin 6 (*IL-6*) and interleukin 8 (*IL-8*) as well as the pro-apoptotic ratio of Bcl-2associated X protein / anti-apoptotic B-cell lymphoma 2 (*BAX/BCL-2*) and protein levels of Cleaved Caspase-3 (17 + 15 KDa), were significantly increased in trophoblasts exposed for 24 h to hypoxia compared to the control condition (S4C–S4E Fig).
(TIF)

**S4 Fig. Increased inflammation and apoptosis in trophoblasts upon hypoxia.** mRNA transcript levels of *TNF-α*, *IL-6* and *IL-8* (**A**) and ratio of *BAX/BCL2* transcript levels (**B**) assessed in placental villous explants exposed to hypoxia or normoxia (control) for 3 h (n = 3) and mRNA transcript levels of *TNF-α*, *IL-6* and *IL-8* (**C**) and ratio of *BAX/BCL2* transcript levels (**D**) and protein levels of Cleaved Caspase-3 (17 + 15 KDa) (**E**) assessed in trophoblasts

exposed to normoxia (control) or hypoxia for 24 h (n = 3-6/experimental condition (n = 3 experiments)). Data are presented as fold change compared to the control and as mean with SEM. $^*$p $\leq$ 0.05, $^{**}$p $\leq$ 0.01 and $^{***}$p $\leq$ 0.001. TNF-α: Tumor necrosis factor α, IL-6: Interleukin 6, IL-8: Interleukin 8, BAX: pro-apoptotic Bcl-2-associated X protein and BCL: anti-apoptotic B-cell lymphoma 2. In contrast to the increased reliance on glycolysis in placental cells upon hypoxia, no differences were observed in activity and mRNA expression levels of the rate-limiting enzyme HADH of the FAO pathway in both placental villous explants and trophoblasts exposed to hypoxia compared to the control group (S5A–S5D Fig).
(TIF)

**S5 Fig. Mitochondrial FAO unaltered in placental cells upon hypoxia.** HADH enzyme activity (**A**) and mRNA transcript levels of *HADH* (**B**) assessed in placental villous explants exposed to hypoxia or normoxia (control) for 3 h (n = 3) and HADH enzyme activity (**C**) and mRNA transcript levels of *HADH* (**D**) assessed in trophoblasts exposed to normoxia (control) or hypoxia for 24 h (n = 3, 6 and 6/experimental condition (n = 3 experiments)). Data are presented as fold change compared to the control and as mean with SEM. HADH: 3-hydroxyacyl-CoA dehydrogenase. Protein levels of nuclear-encoded sub-units of the ETC including Ndufb8, Sdhb, UQCR2 and ATP5A of respectively complex I, II, III and V, were not affected in placental villous explants exposed for 3 h to hypoxia (S6A Fig). Also, related mRNA transcript levels of *Ndufb3*, *Cycl-1*, *COXIV* and *COXII* of respectively complex I, III, IV and IV, were not affected in placental villous explants after exposure to hypoxia (S6B Fig). Protein levels of UQCRC2 (complex III) as well as mRNA transcript levels of all investigated sub-units including mitochondrial encoded *COXII* were significantly decreased in trophoblasts upon 24h exposure to hypoxia (S6D–S6F Fig).
(TIF)

**S6 Fig. Alterations in nuclear-encoded OXPHOS sub-units in trophoblasts upon hypoxia.** Protein levels of nuclear-encoded OXPHOS sub-units (Ndufb8, Sdhb, UQCRC2 and ATP5A respectively Complex I, II, III and V) (**A**) and mRNA transcript levels of nuclear-encoded OXPHOS sub-units (*Ndufb*, *Cycl-1*, *COXIV* and COXII respectively Complex I, III, IV and IV) (**B**) assessed in placental villous explants exposed to normoxia (control) or hypoxia for 3 h (n = 3) and protein levels of nuclear-encoded OXPHOS sub-units (Ndufb8, Sdhb, UQCRC2 and ATP5A respectively Complex I, II, III and V) (**C**) and mRNA transcript levels of nuclear-encoded OXPHOS sub-units (*Ndufb*, *Cycl-1*, *COXIV* and COXII respectively Complex I, III, IV and IV) (**D**) assessed in trophoblasts exposed to normoxia (control) or hypoxia for 24 h (n = 3-6/experimental condition (n = 3 experiments)). Representative immunoblots are shown and Western blots were corrected for total protein loading assessed by Ponceau S Staining with adjusted contrast equally applied to the whole photograph. Black boxes around the representative pictures indicate that they were cut from the same Western blot. Data are presented as fold change compared to the control and as mean with SEM. Ndufb3: NADH dehydrogenase [ubiquinone] 1 beta subcomplex subunit 3, Sdhb: Succinate dehydrogenase [ubiquinone] iron-sulfur subunit, UQCRC2: Cytochrome b-c1 complex subunit 2, ATP5A: ATP synthase F1 subunit alpha, Cyc1: Cytochrome C1, COXIV: Cytochrome *c* oxidase subunit IV and COXII: Cytochrome *c* oxidase subunit II. All investigated constituents of the autophagy in our study were unaltered in placental villous explants exposed for 3h to hypoxia compared to the control condition (S7A and S7B Fig). In cultured trophoblasts, 24 h hypoxia resulted in a significant decrease in autophagy-associated LC3BI and LC3BII protein levels while the ratio of LC3BI/LC3BII was significantly increased compared to the normoxic condition. In addition, while both protein as well as mRNA levels of GABA Type A Receptor Associated Protein Like 1 (*GABARAPL1*) were increased in response to hypoxia, protein levels of SQSTM1 were

decreased and its transcript levels increased upon hypoxia (S7C and S7D Fig). *OPTN* mRNA levels increased 2-fold in response to hypoxia (S7D Fig).
(TIF)

**S7 Fig. Expression of key constituents of the autophagy machinery is altered in trophoblast upon hypoxia.** Autophagy-associated protein levels of SQSTM1, GABARAPL1, LC3BI, LC3BII and LC3BII / LC3BI ratio (**A**) and mRNA transcript levels of *SQSTM1*, *GABARAPL1*, *OPTN and LC3B* (**B**) assessed in placental villous explants exposed to normoxia (control) or hypoxia for 3 h (n = 3) and autophagy-associated protein levels of SQSTM1, GABARAPL1, LC3BI and LC3BII (**C**) and mRNA transcript levels of *SQSTM1*, *GABARAPL1*, *OPTN and MAP1LC3A/B* (**D**) assessed in trophoblasts exposed to normoxia (control) or hypoxia for 24 h (n = 3-6/experimental condition (n = 3 experiments)). Representative immunoblots are shown and Western blots were corrected for total protein loading assessed by Ponceau S Staining with adjusted contrast equally applied to the whole photograph. Black boxes around the representative pictures indicate that they were cut from the same Western blot. Data are presented as fold change compared to the control and as mean with SEM. $^{*}p \leq 0.05$ and $^{***}p \leq 0.001$. SQSTM1: Sequestosome 1, GABARAPL1: GABA Type A Receptor Associated Protein Like 1, LC3B: Microtubule-associated protein 1 light chain 3 beta, OPTN: Optineurin and LC3A: Microtubule-associated protein 1 light chain 3 alpha.
(TIF)

**S1 Table. Primers used for qPCR.**
(DOCX)

## Author Contributions

**Conceptualization:** Philippe Vangrieken, Alex H. V. Remels.

**Data curation:** Philippe Vangrieken, Pieter A. Leermakers, Christy B. M. Tulen.

**Formal analysis:** Philippe Vangrieken, Pieter A. Leermakers, Ger. M. J. Janssen, Iris Kaminski, Iris Geomini, Titus Lemmens.

**Project administration:** Christy B. M. Tulen.

**Supervision:** Salwan Al-Nasiry, Aalt Bast, Paul M. H. Schiffers, Frederik J. van Schooten, Alex H. V. Remels.

**Visualization:** Philippe Vangrieken.

**Writing – original draft:** Philippe Vangrieken, Alex H. V. Remels.

**Writing – review & editing:** Philippe Vangrieken, Salwan Al-Nasiry, Aalt Bast, Paul M. H. Schiffers, Frederik J. van Schooten, Alex H. V. Remels.

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
