## [Decision Letter · Decision Letter 0]

25 Mar 2020

PONE-D-20-05832

Hypoxia-induced mitochondrial abnormalities in cells of the placenta

PLOS ONE

Dear Mr Vangrieken,

Thank you for submitting your manuscript to PLOS ONE. After careful consideration, we feel that it has merit but does not fully meet PLOS ONE’s publication criteria as it currently stands. Therefore, we invite you to submit a revised version of the manuscript that addresses the points raised during the review process.

Your manuscript was reviewed by two experts  and both of them suggested major experiments to improve your manuscript. 

We would appreciate receiving your revised manuscript by May 09 2020 11:59PM. To enhance the reproducibility of your results, we recommend that if applicable you deposit your laboratory protocols in protocols.io, where a protocol can be assigned its own identifier (DOI) such that it can be cited independently in the future. For instructions see: http://journals.plos.org/plosone/s/submission-guidelines#loc-laboratory-protocols

We look forward to receiving your revised manuscript.

Kind regards,

Partha Mukhopadhyay, Ph.D.

Academic Editor

PLOS ONE

Journal Requirements:

2. Thank you for specifying in your ethics statement that participant consent was verbal. Please also specify: 1) whether the ethics committee approved the verbal consent procedure, 2) why written consent could not be obtained, and 3) how verbal consent was recorded.

"This study was supported by NUTRIM Graduate Program. There was no additional external funding received for this study."

Reviewers' comments:

Reviewer's Responses to Questions

**Comments to the Author**

1. Is the manuscript technically sound, and do the data support the conclusions?

Reviewer #1: Partly

Reviewer #2: Partly

2. Has the statistical analysis been performed appropriately and rigorously? 

Reviewer #1: Yes

Reviewer #2: Yes

3. Have the authors made all data underlying the findings in their manuscript fully available?

Reviewer #1: Yes

Reviewer #2: Yes

4. Is the manuscript presented in an intelligible fashion and written in standard English?

Reviewer #1: Yes

Reviewer #2: Yes

5. Review Comments to the Author

Reviewer #1: In the current study "Hypoxia-induced mitochondrial abnormalities in cells of the placenta" the authors analyzed the mitochondrial features in placenta explants and a trphoblast cell line under hypoxia. Here are some major concerns:

1. The authors provided evidence that the hypoxic trophoblast cells have higher molecular capacity of glycolysis, which is a common phenotype in most cells under hypoxia. Have the authors analyzed the metabolism pattern of the cells with oxygen consumption, pH change and glycolysis product quantification? How is glycolysis linked to the onset of PE?

2. The authors want to link the dysfunction of mitochondria with cell apoptosis and mitophagy. However, gene expression analysis with mRNA and western blot protein quantification is not enough to make a conclusion. The authors need to analyze apoptosis with TUNEL staining and caspase-3 cleavage. Also, the authors need to confirm mitophagy with electromicroscopy.

3. The authors tried to analyze the biogenesis, fission and fusion of mitochondria and provide explanation for the dropped number of mitochondria. All these gene expression results need to be paired with morphological study with electromicroscopy to make a conclusion. Also, the morphological mitochondrial changes and damages can be seen in such experiments.

4. The authors should analyze the mitochondrial membrane potential.

Reviewer #2: The authors have done an incredible job of pursuing and presenting their work effectively. The rationale, experimental design, and results are convincing. However, the manuscript presents specific gaps addressing which will improve this work substantially.

Major

1. I like that the authors have provided extensive details of the experimental techniques and assay procedures that they used in the project. However, I would like to point out that the table consisting of the list of primary antibodies should be part of the main manuscript, not a supplemental table. The table should provide details of the exact dilutions used for the specific primary antibodies used in the western blots. Currently, the table lacks the dilutions for each antibody.

2. Throughout the manuscript, the authors have used a 3-hr window for subjecting the placental explants to hypoxic stress. They do not provide any explanation on the rationale until we reach the discussion (lines 465-467) about the selected number of hours. First, the authors should give the justifications of their experimental conditions at the beginning – in the relevant results section before stating the precise outcomes of the experiments. Second, they should consider providing experimental evidence of placental tissue deterioration rather than only providing references. They should consider presenting GSSG/GSH, and TEAC fold changes when the placental explants are subjected to hypoxic stress over 2, 4, 5, and/or 7 hours, along with 3 hours, to reveal the changes observed with these parameters, concomitant with reported tissue deterioration. This data will provide experimental rigor to their findings and will substantiate their scientific decision to choose 3 hours for placental explants.

3. In Fig 2, the authors present the effect of hypoxic stress on mtDNA copy number and citrate synthase (CS) enzyme activity on the placental explants and BeWo cells. However, they have presented the effect of mitochondrial antioxidants MitoQ and Quercetin only on BeWo cells. Have the authors performed the rescue experiment on the placental explants too? These results will consolidate their findings with the cultured trophoblasts. The authors should consider presenting the data as part of Fig 2.

4. There are significant discrepancies in results between the placental explants and the BeWo cell line. The authors have not explained each sub-figure of every figure in detail. Also, the discussion part of the manuscript hardly refers back to specific figures and results while providing the rationales of their overall findings. The discussion has to be streamlined with references to particular figures with explanations to every data obtained in them.

Minor

1. Figs 2 through 6 and supplemental Figs 4 and 5 present western blots that are lacking the molecular weight markers for each panel, they should be included in the respective figures.

2. The authors mention that they have used the Ponceau Staining image for normalization of the band signal of each protein on the blots, so, they should consider including the Ponceau stained membrane image in the figures.

6. PLOS authors have the option to publish the peer review history of their article (what does this mean?). If published, this will include your full peer review and any attached files.

Reviewer #1: No

Reviewer #2: No

---

## [Author Response · Author response to Decision Letter 0]

10 Nov 2020

See document "Response to Reviewers"

---

## [Decision Letter · Decision Letter 1]

23 Dec 2020

Hypoxia-induced mitochondrial abnormalities in cells of the placenta

PONE-D-20-05832R1

Dear Dr. Vangrieken,

We’re pleased to inform you that your manuscript has been judged scientifically suitable for publication and will be formally accepted for publication once it meets all outstanding technical requirements.

Kind regards,

Partha Mukhopadhyay, Ph.D.

Section Editor

PLOS ONE

Additional Editor Comments (optional):

Reviewers' comments:

Reviewer's Responses to Questions

**Comments to the Author**

1. If the authors have adequately addressed your comments raised in a previous round of review and you feel that this manuscript is now acceptable for publication, you may indicate that here to bypass the “Comments to the Author” section, enter your conflict of interest statement in the “Confidential to Editor” section, and submit your "Accept" recommendation.

Reviewer #1: All comments have been addressed

2. Is the manuscript technically sound, and do the data support the conclusions?

Reviewer #1: Yes

3. Has the statistical analysis been performed appropriately and rigorously? 

Reviewer #1: Yes

4. Have the authors made all data underlying the findings in their manuscript fully available?

Reviewer #1: Yes

5. Is the manuscript presented in an intelligible fashion and written in standard English?

Reviewer #1: Yes

6. Review Comments to the Author

Reviewer #1: (No Response)

7. PLOS authors have the option to publish the peer review history of their article (what does this mean?). If published, this will include your full peer review and any attached files.

Reviewer #1: No

---

## [Editor Report · Acceptance letter]

2 Jan 2021

PONE-D-20-05832R1 

Hypoxia-induced mitochondrial abnormalities in cells of the placenta 

Dear Dr. Vangrieken:

I'm pleased to inform you that your manuscript has been deemed suitable for publication in PLOS ONE. Congratulations! Your manuscript is now with our production department. 

Kind regards, 

on behalf of

Dr. Partha Mukhopadhyay 

Section Editor

PLOS ONE